# Probing the Influence of Single-Site Mutations in the Central Cross-β Region of Amyloid β (1–40) Peptides

**DOI:** 10.3390/biom11121848

**Published:** 2021-12-09

**Authors:** Jacob Fritzsch, Alexander Korn, Dayana Surendran, Martin Krueger, Holger A. Scheidt, Kaustubh R. Mote, Perunthiruthy K. Madhu, Sudipta Maiti, Daniel Huster

**Affiliations:** 1Institute for Medical Physics and Biophysics, Leipzig University, Härtelstr. 16–18, D-04107 Leipzig, Germany; jacobfritzsch@gmail.com (J.F.); alexander.korn@medizin.uni-leipzig.de (A.K.); holger.scheidt@medizin.uni-leipzig.de (H.A.S.); 2Department of Chemical Sciences, Tata Institute of Fundamental Research, Homi Bhabha Road, Colaba, Mumbai 400005, India; dayana.surendran@gmail.com (D.S.); sudipta.maiti@gmail.com (S.M.); 3Institute of Anatomy, Leipzig University, Liebigstr. 13, D-04103 Leipzig, Germany; martin.krueger@medizin.uni-leipzig.de; 4Tata Institute of Fundamental Research Hyderabad, 36/P Gopanpally Village, Serilingampally Mandal, Hyderabad 500046, India; krmote@tifrh.res.in (K.R.M.); madhu@tifr.res.in (P.K.M.)

**Keywords:** solid-state NMR, amyloid structure formation, fibril dynamics

## Abstract

Amyloid β (Aβ) is a peptide known to form amyloid fibrils in the brain of patients suffering from Alzheimer’s disease. A complete mechanistic understanding how Aβ peptides form neurotoxic assemblies and how they kill neurons has not yet been achieved. Previous analysis of various Aβ_40_ mutants could reveal the significant importance of the hydrophobic contact between the residues Phe_19_ and Leu_34_ for cell toxicity. For some mutations at Phe_19_, toxicity was completely abolished. In the current study, we assessed if perturbations introduced by mutations in the direct proximity of the Phe_19_/Leu_34_ contact would have similar relevance for the fibrillation kinetics, structure, dynamics and toxicity of the Aβ assemblies. To this end, we rationally modified positions Phe_20_ or Gly_33_. A small library of Aβ_40_ peptides with Phe_20_ mutated to Lys, Tyr or the non-proteinogenic cyclohexylalanine (Cha) or Gly_33_ mutated to Ala was synthesized. We used electron microscopy, circular dichroism, X-ray diffraction, solid-state NMR spectroscopy, ThT fluorescence and MTT cell toxicity assays to comprehensively investigate the physicochemical properties of the Aβ fibrils formed by the modified peptides as well as toxicity to a neuronal cell line. Single mutations of either Phe_20_ or Gly_33_ led to relatively drastic alterations in the Aβ fibrillation kinetics but left the global, as well as the local structure, of the fibrils largely unchanged. Furthermore, the introduced perturbations caused a severe decrease or loss of cell toxicity compared to wildtype Aβ_40_. We suggest that perturbations at position Phe_20_ and Gly_33_ affect the fibrillation pathway of Aβ_40_ and, thereby, influence the especially toxic oligomeric species manifesting so that the region around the Phe_19_/Leu_34_ hydrophobic contact provides a promising site for the design of small molecules interfering with the Aβ fibrillation pathway.

## 1. Introduction

Fibrillation of the amyloid β (Aβ) peptide is considered the most important molecular hallmark of Alzheimer’s disease (AD) [1]. Developing small molecules or antibodies that alter or inhibit this fibrillation process in a solution has largely failed in the last decade. Until now, there is only one drug available which was recently FDA-approved with a somewhat controversial clinical outcome [2]. One of the reasons for the highly unsatisfactory situation is that our understanding of the structural characteristics of the Aβ oligomer and fibril formation and how it causes neuronal toxicity remains rather poor. In particular, the structural intermediates evolving during the aggregation process which have been assessed as being the key malign species are highly transient and diversified rendering them a scientific subject that features even more obscurity and necessitates further investigation and research. Any structural information about the Aβ peptide conversion into oligomers and finally fibrils is considered crucial for developing new strategies for tackling AD [3,4,5].

A key point for the understanding of the molecular details of amyloid formation from individual peptides is to disclose the role of individual amino acids and local molecular contacts between the amino acid side chains for the peptide’s structure formation properties and especially its toxicity. One approach addressing this question is to rationally modify the peptide sequence via single mutations to cause local physical perturbations to the peptide structure [6,7,8,9,10]. This way, the various electrostatic and hydrophobic forces that control the fibrillation pathway and entail the amyloid stability are altered and the consequential effect of single mutations on important Aβ peptide characteristics can be investigated. Critical amino acids as well as structurally important regions and contacts within the structure of the peptide assembly can be defined and characterized.

We have recently systematically investigated Aβ_40_ peptide mutants that carried alterations in the most often described hydrophobic contact between Phe_19_ and Leu_34_ [11,12,13,14,15,16,17], which allowed for comparing the fibrillation kinetics, the global structure and morphology, the local structure near the mutation site and the cell toxicity of mutated Aβ peptides with wildtype (WT) Aβ_40_. With these models, we could reveal the high significance of the hydrophobic contact between Phe_19_ and Leu_34_ for the formation of Aβ_40_ oligomers and fibrils. The importance of Phe_19_ for the formation of a lowly populated disordered 3–10 helix was also demonstrated [18].

Single mutations of Phe_19_ to other natural amino acids led to significant alterations in the fibrillation kinetics, while leaving the morphology of the fibrils largely unchanged [11,12,13,14,15,16,17]. However, the secondary structure and local dynamics in the vicinity of the mutated site changed rather drastically for some of the mutants [14]. The most important result was that rather conservative mutations *completely* abolished the toxicity of the Aβ_40_ species [13]. We then mutated the residue Leu_34_ to *D*-Leu, Ile or Val, all representing very minor modifications and, consequently, observed no structural or morphological alterations of the fibrils and toxicity values close to WT [15]. These experiments suggested that Phe_19_ may carry significant importance in the formation of toxic oligomers. This hypothesis was further probed by replacing Phe_19_ by non-natural amino acids with varying ring structures [16]. These molecules showed very accelerated fibrillation kinetics but largely attenuated toxicity supporting the hypothesis that Aβ_40_ toxicity is highly dependent on the residue Phe_19_.

Taken together, the aforementioned results suggest that the toxicity of Aβ_40_ may strongly depend on the early folding contact between Phe_19_ and Leu_34_. In the current study, this hypothesis was challenged. In particular, we asked the question as to how specific the effects of mutations of Phe_19_ and Leu_34_ are to these amino acids and if similar effects are also obtained if two neighboring residues in position 20 (Phe_20_) or 33 (Gly_33_) were mutated. To this end, we created a small library of Aβ_40_ peptide mutants where Phe_20_ or Gly_33_ were replaced by residues that showed the most drastic effects in the previous study [11,12,16].

Both amino acids are located in direct proximity to the Phe_19_/Leu_34_ contact but the Phe_20_ side chain points out of the fibril core according to the current models of Aβ_40_ [19,20,21,22]. It should be mentioned that there is also one model describing the Phe_20_ side chain to point into the fibrillar core [21]. Any side chain that is introduced into position 33 should also point outwards. The residues chosen to replace Phe_20_ were (i) Lys, (ii) Tyr and (iii) the non-proteinogenic cyclohexylalanine (Cha). With this choice of side chains, we were able to first study the effect of electrostatics by introducing the positively charged Lys; second, the possibility to adopt an altered hydrogen bond pattern by introducing aromatic Tyr; third, the role of π stacking and aromaticity by mutation to cyclohexylalanine (Cha) where the aromatic phenyl ring is replaced by the saturated cyclohexyl moiety. The mutation introduced to replace Gly_33_ was Ala, which represents the most conservative mutation in this position. Therefore, we aimed to obtain a detailed view on the role of this amino acid within the structure of the Aβ_40_ peptide assembly to assess if it might be worth further investigations.

We used fluorescence spectroscopy to monitor fibrillation kinetics, X-ray diffraction and circular dichroism (CD) spectroscopy to investigate the effect of the alterations in local physical forces on the cross-β structure as well as the global secondary structure and transmission electron microscopy (TEM) to examine the fibril morphology in response to the mutational changes. The consequences of these modifications on the local structure and dynamics were probed by solid-state NMR spectroscopy. Finally, standard MTT assays were carried out to determine the influence of these modifications on the cellular toxicity of the peptides.

## 2. Materials and Methods

### 2.1. Peptides Synthesis

WT Aβ_40_ peptides (DAEFRHDSGY EVHHQKLVF**F** AEDVGSNKGA II**G**LMVGGVV) and Aβ_40_ mutants were synthesized using standard F-moc solid phase synthesis. The purity was >98%, tested by HPLC and MS-spectroscopy. Four Aβ_40_ peptides with mutations at either Phe_20_ or Gly_33_ (indicated in bold in the sequence above) were synthesized with uniformly ^13^C/^15^N-labeled amino acids (Euriso-TOP; Saarbrücken, Germany) at position Val_18_, Phe_19_ and Ala_2__1_ (underlined in the sequence above). In three mutants, Phe_20_ was replaced by either Lys, Tyr or cyclohexylalanine (Cha). In the fourth mutant, residue Gly_33_ was replaced by Ala. The mutated amino acids are indicated with bold letters in the sequence above. An alternatively labeled peptide version of the fourth mutant was synthesized with uniformly ^13^C/^15^N-labeled amino acids at position Phe_19_, Asp_23_, Gly_25_, Leu_34_ and uniformly ^15^N-labeled amino acid at position Lys_28_.

### 2.2. Sample Preparation for NMR Spectroscopy, X-ray Diffraction Measurements and Transmission Electron Microscopy of Aβ Fibrils

Lyophilized peptides were dissolved in aqueous buffer containing 25 mM of sodium phosphate, 150 mM of sodium chloride and 0.01% of sodium azide at a concentration of 1 mg/mL at pH 9.2. Directly after complete solubilization, the peptide solutions were dialyzed against the same buffer at pH 7.4 for 4 h with a buffer exchange after 2 h. For dialysis, 1000 Da molecular weight cut-off dialyses tubes (ZelluTrans/ROTH; Karlsruhe, Germany) were used. The peptide solutions were then transferred into reaction tubes and incubated at 37 °C for ≥7 days while continuously shaken at 450 rpm (Thermomixer comfort, Eppendorf; Köln/Wesseling, Germany).

For NMR measurements, fibril solutions were centrifuged at 81,500× *g* for 2 h at 4 °C. After removal of the supernatant, pellets were lyophilized for at least 72 h, rehydrated to 50 wt% H_2_O, homogenized by freezing in liquid nitrogen and thawed at 37 °C ten times and transferred into 3.2 mm MAS rotors and stored at −24 °C.

For X-ray diffraction measurements fibril solutions were centrifuged at 15,060× *g* for 10 s and washed with H_2_O three times. Approximately 10 µL of highly concentrated fibril solution was pipetted between two paraffin coated glass capillaries and slowly dried in a desiccator to give the fibrils time to collimate.

For transmission electron microscopy, fibril solutions were diluted 1:20 with H_2_O, then 2 µL were applied onto a formvar film-coated copper grid, dried for 30 min and subsequently stained with 1% uranyl acetate in pure water.

### 2.3. Sample Preparation for NMR Spectroscopy of Aβ Oligomers

For Aβ oligomer preparation [23], approximately 5 mg of lyophilized peptides were dissolved in 1 mL of ammonia solution at pH 11. Aqueous buffer containing 175 mM of ammonium acetate (pH 7.4) was added to a final peptide concentration of 0.17 mg/mL. The peptide solution was incubated for 30 min at room temperature without shaking and then dripped into liquid nitrogen to stop the oligomerization process and prevent fibrillation. The flash-frozen sample was lyophilized for 4 days to remove the water and the buffer. The obtained powder was transferred into a 3.2 mm MAS rotor and stored at −24 °C.

### 2.4. Fibrillation Kinetics Measurements by ThT-Fluorescence Assay

For recording the Aβ_40_ fibrillation kinetics, Thioflavin T (ThT) was used as a fluorescence dye which shows an increased quantum yield at 482 nm under excitation at 440 nm when bound to cross-β structures, exhibited by mature Aβ fibrils [24]. Lyophilized peptides were first dissolved in dimethyl sulfoxide (DMSO) (30 µg peptide/µL DMSO). The DMSO-peptide solution was then diluted with buffer containing 25 mM of sodium phosphate, 150 mM of sodium chloride, 0.01% of sodium azide and 20 µM of ThT at pH 7.4 to a final peptide concentration of 30 µM. Fluorescence spectra were recorded using a Tecan Infinite M200 microplate reader (Tecan Group AG; Männedorf, Switzerland) and a 96-well plate format. For each measurement, 100 µL of the peptide solution were incubated at 37 °C under an applied shaking cycle of 2 s shaking (2 mm shaking amplitude and 579.8 rpm) followed by a 5 min waiting period. The fluorescence intensity was measured every 5 min. Empty wells were filled with H_2_O to prevent dehydration of the sample. Fluorescence intensity measurements were carried out in three independent measurements in quintuplicate for each peptide. Intensity data were normalized for each sample so that the values ranged between 0 and 1. The data were fitted using a sigmoidal curve according to
(1)$$I=y_{i}+m_{i}t+\frac{y_{f}+m_{f}t}{1+e^{-\left[\left(t-t_{0}\right)/\tau \right]}}$$
where *I* is the normalized fluorescence intensity, *t* the time, *t*_0_ the characteristic time, where half the maximal intensity is reached, *m* the slope of the graph in lag time ($m_{i}$) and plateau phase ($m_{f}$). *τ* represents a measure of the fibrillation time from which the lag time is calculated by *t_lag_* = *t*_0_ − 2 × *τ* and the fibrillation time is calculated by *t_fib_* = 4 × *τ* [25].

### 2.5. Circular Dichroism Spectroscopy

Lyophilized peptides were dissolved in buffer containing 25 mM of sodium phosphate, 150 mM of sodium chloride and 0.01% of sodium azide at a concentration of 1 mg/mL at pH 7.4. Aliquots of the peptide solutions were used either directly for CD measurements in the beginning of the fibrillation period or after ≥13 days of incubation at 37 °C and 450 rpm. Aliquots were diluted with H_2_O for CD measurements. Reported values represent the mean of two independent measurements with a final peptide concentration of 20 μM and 10 μM. CD spectra were recorded in a 2 mm quartz cuvette on a Jasco J-710 spectropolarimeter at 25 °C. All measurements were carried out fivefold.

### 2.6. Transmission Electron Microscopy (TEM)

TEM-Images were obtained using a Zeiss SIGMA microscope equipped with a STEM detector and Atlas Software (Zeiss NTS; Oberkochen, Germany).

### 2.7. X-ray Diffraction Measurements

Measurements were carried out using a copper rotating anode MM007 with 0.8 kW as the goniometer head and R-AXIS IV++ as the image plate detector (Rigaku; Tokyo, Japan). The exposure time was 3 min at 24 °C.

### 2.8. Solid-State NMR Spectroscopy

NMR spectra were acquired using a Bruker Avance III 600 MHz NMR spectrometer (Bruker BioSpin GmbH; Rheinstetten, Germany) operating at a resonance frequency of 60.8 MHz for ^15^N, 150.9 MHz for ^13^C and 600.1 MHz for ^1^H or a Bruker Avance Neo 700 MHz NMR spectrometer (Bruker Biospin GmbH; Rheinstetten, Germany) operating at a resonance frequency of 70.9 MHz for ^15^N, 176.1 MHz for ^13^C and 700.1 MHz for ^1^H both under magic-angle spinning (MAS). A triple channel 3.2 mm MAS probe was used. MAS frequencies were 5 kHz for DIPSHIFT experiments and 11.777 kHz for all other experiments. The temperature was set to 30 °C. The pulse lengths of the 90° pulses were set to 4 µs for ^1^H, 4 µs for ^13^C and 5 µs or 6 µs for ^15^N. ^1^H-X CP contact times were 700 µs for DIPSHIFT experiments and 1000 µs for all other experiments at a spin lock field of approximately 50 kHz. The relaxation delay was 2.5 s. ^1^H dipolar decoupling was applied with a rf amplitude of 65 kHz using SPINAL64 during acquisition.

Two dimensional ^13^C-^13^C DARR experiments and ^13^C-^15^N correlation spectra were performed simultaneously using dual-acquisition [26]. The DARR mixing time was 500 ms. Alongside each DARR experiment, 4 or 8 ^13^C-^15^N correlation spectra were acquired with a CP contact time of 4 ms for ^13^C-^15^N transfer.

^1^H-^13^C dipolar coupling measurements were carried out using constant time DIPSHIFT experiments [27]. The frequency switched Lee–Goldburg sequence (FSLG) [28] with an effective rf field of 80 kHz was applied for homonuclear decoupling during dipolar evolution. The strength of the dipolar coupling was obtained from numerical simulations of the dephasing curves for each resolved carbon atom. Order parameters were calculated by dividing the results by reference values for the rigid limits [29].

^13^C-^15^N frequency selective REDOR experiments [30] were conducted on a Bruker Avance III 700 MHz spectrometer at an MAS frequency of 8.333 kHz. ^13^C *π* pulses were 10.5 µs and the carbonyl selective pulse was 500 µs long. Δ*S*/*S* curves were simulated assuming only an isolated ^13^C-^15^N contact.

### 2.9. 3-(4,5-Dimethylthiazol-2-yl)-2,5-diphenyltetrazoliumbromid (MTT) Assay

The standard MTT assay was carried out to determine the viability of cells treated with the Aβ (1–40) peptide variants. RN46A cells were seeded into 96-well plates at a density of 1 × 10^4^ cells/well and cultivated for 24 h. After attachment, the cells were treated with either the vehicle control or the Aβ (1–40) WT and mutant species at 100 µM for 60 h at 37 °C. Subsequently, MTT solution (1 mg/mL in PBS) was added into each well and cells were incubated at 37 °C for 4 h. The MTT solution was aspirated and 100 µL of acidified isopropanol was added to each well. The plate was gently shaken on an orbital shaker for 20 min to completely dissolve the formazan precipitation. Absorbance was detected at 570 nm using a microplate reader (TECAN Infinite M200, Redmond, WA, USA). Absorbance was normalized with respect to the untreated control cultures to calculate changes in cell viability. Three replicates were prepared, i.e., three different experiments, with six wells each for each peptide.

## 3. Results

### 3.1. Fibrillation Kinetics

The fibrillation kinetics of all Aβ_40_ peptide variants were studied using the standard Thioflavin T (ThT) fluorescence assay. A plot of the normalized fluorescence intensity of WT and mutated Aβ_40_ peptides as a function of time is given in Figure 1. The Gly_33_Ala, Phe_20_Tyr and Phe_20_Lys mutants showed faster fibrillation kinetics compared to WT Aβ_40_. For the Phe_20_Cha variant slower fibrillation kinetics were observed.

In a three-step model, amyloid fibril formation can be described as a transient process that starts with a lag phase followed by a fibrillation phase, which equilibrates to the plateau phase. While the lag phase represents early forms of organization of the Aβ peptide, the growth of the fibrils typically leads to a steeply increasing curve progression described as sigmoidal increase caused by the enhancement of the fibrillation process due to already emerged seeds and early fibrils.

From the measurements, we obtained two kinetic parameters that characterize the fibrillation process: the lag time corresponding to the length of the lag phase and the characteristic time indicating the time required to reach half the maximal ThT fluorescence intensity. For WT Aβ_40_ under our conditions, we observed a lag time of 7.3 ± 2.4 h and a characteristic time of 9.0 ± 2.0 h. The Gly_33_Ala and Phe_20_Tyr mutants exhibited significantly decreased lag and characteristic times compared to WT Aβ_40_ with the strongest decrease in Gly_33_Ala (*t_lag_*~0; *t_char_* = 1.5 ± 0.5 h), followed by Phe_20_Tyr (*t_lag_* = 3.8 ± 1.3 h; *t_char_* = 5.3 ± 1.2 h). The Phe_20_Lys mutant (*t_lag_* = 4.9 ± 1.0 h; *t_char_* = 7.3 ± 0.8 h) showed accelerated fibrillation kinetics as well. Only the Phe_20_Cha mutant (*t_lag_* = 13.5 ± 3.1 h; *t_char_* = 14.9 ± 2.8 h) featured a significantly decelerated fibrillation kinetics compared to WT (*p* < 0.005%). We detected the typical sigmoidal increase in ThT fluorescence intensity for the WT Aβ peptide as well as the variants Phe_20_Tyr, Phe_20_Lys and Phe_20_Cha. In the Gly_33_Ala mutant, no lag time and no sigmoidal increase in ThT fluorescence intensity was observed in our assay which comprised a dead time of approximately five minutes before the first measurement was performed.

### 3.2. Transmission Electron Microscopy

Transmission electron microscopy (TEM) was used to study the general fibril morphology. Electron micrographs of all variants are shown in Figure 2A–E. The TEM images confirm that WT Aβ_40_ peptides as well as all mutants formed fibrils of comparable morphology. The Phe_20_Cha and Phe_20_Lys mutants showed similar morphology as the WT while the fibrils of Phe_20_Tyr appeared somehow longer and arranged themselves in a more parallel manner. This is in contrast to the Gly_33_Ala mutant fibrils, which appeared shorter and overlapped more. We determined the average fibril diameters of all variants from the TEM images. Fibril diameters of the WT (10.4 ± 1.1 nm) and Gly_33_Ala variant (11.5 ± 1.4 nm) were almost equal. All three studied mutations at Phe_20_ led to an increase in the diameter (Phe_20_Cha: 13.6 ± 1.6 nm; Phe_20_Tyr: 14.3 ± 1.1 nm; Phe_20_Lys: 15.5 ± 1.6 nm).

### 3.3. X-ray Diffraction

A more detailed view on the global structure of the mutated Aβ_40_ fibrils was provided by X-ray diffraction. X-ray diffraction data of all Aβ_40_ peptides is presented in Figure 2F–J. All Aβ mutants featured similar XRD spectra as the WT peptide and prove that the cross-β structure of amyloids, which is formed by two stacked β-sheets, is formed by all mutated peptides. In particular, the spectra show a strong meridional reflection at ~4.7 Å, indicating the intrasheet distance between the neighboring β-strands corresponding to the hydrogen bond length and a broader equatorial reflection from ~9.5 to 10.3 Å indicating the distance between the two opposing β-strands. The meridional reflection in all variants was well-defined at 4.6 or 4.7 Å in agreement with the regular β-sheet structure of the WT and mutated fibrils. The more diffuse equatorial reflection was observed in WT fibrils at approximately 9.5 Å and in the mutants at 9.8 Å (Phe_20_Lys), 10.0 Å (Phe_20_Cha; Gly_33_Ala) and 10.3 Å (Phe_20_Tyr).

The X-ray diffraction data reveals that the global structure of the WT Aβ_40_ fibrils prevailed throughout the mutants and that the cross-β structure of Aβ_40_ is very resistant against local perturbations introduced by our single mutations.

### 3.4. Circular Dichroism Spectroscopy

CD measurements were carried out at two different time points to obtain information about the secondary structure directly after diluting the peptides in the phosphate buffer and after a fibrillation time of 13 or more days under standard conditions. The freshly dissolved peptides all exhibited a strong negative peak at ~197 nm, characteristic for random coil structure in the disordered peptides (Figure 3A). After incubation, the spectra of all variants changed drastically, featuring a minimum at ~216–227 nm and a maximum at ~196 nm (Figure 3B). This indicated a transition to β-sheet secondary structure [31].

### 3.5. Local Structure of the Aβ_40_ Fibrils by ^13^C NMR Chemical Shift Measurements

To examine the local structure of mutated Aβ_40_, peptide variants with ^13^C/^15^N-labels at selected positions (Val_18_, Phe_19_ and Ala_21_) were fibrillated and studied using solid-state NMR spectroscopy. For all samples, we obtained well-resolved one-dimensional ^13^C CP MAS NMR spectra. The corresponding NMR spectra featured relatively well-resolved NMR lines indicating moderate polymorphism in the local structure. We were able to assign all chemical shift values to the labeled carbons from the two-dimensional ^13^C–^13^C correlation NMR spectra (DARR experiment). These chemical shifts represent a measure of the local secondary structure since the dihedral angles of the peptide backbones influenced the magnetic environment of the nuclei, especially the ^13^Cα, ^13^Cβ and ^13^CO position [32]. To be independent of external referencing, ^13^Cα–^13^Cβ chemical shift differences [33] were calculated and are shown in Figure 4. Specific values for the respective amino acids indicating secondary structure motifs are also displayed. As apparent from Figure 4, WT Aβ_40_ fibrils featured well-defined β-sheet structure for the reported amino acids in agreement with the current models for Aβ_40_ fibrils [19,20,21,22]. All mutated variants did not show substantial deviations from the WT Aβ secondary structure for the reported amino acids. All determined values were in agreement with β-sheet structure.

The structural homogeneity of all preparations is particularly obvious at position Val_18_, with values from 25.5–26.1 ppm for the ^13^Cα–^13^Cβ chemical shift differences in all variants. Residues Phe_19_ and Ala_21_, covalently bound to the amino acid in position 20, were mutated in most of the peptides. ^13^Cα–^13^Cβ chemical shift difference values at position Phe_19_ ranged from 12.8 to 14.9 ppm with the largest deviation from the WT (13.7 ppm) for Phe_20_Tyr (14.9 ppm) followed by Gly_33_Ala (14.5 ppm) and Phe_20_Lys (14.3 ppm). At position Ala_21_, very similar chemical shift differences were observed for the mutants with values from 26.9–29.0 ppm for all variants and the largest deviations from the WT (29.0 ppm) for Phe_20_Lys (26.9 ppm) and Phe_20_Tyr (27.0 ppm).

Chemical shift data for all variants and labeled amino acids suggest very homogenous local secondary structures of the various Aβ_40_ fibrils with slight deviations most likely in the Phe_20_Lys and Phe_20_Tyr mutants.

### 3.6. Investigation of the Molecular Dynamics of the Mutated Aβ_40_ Fibrils

In addition to secondary structure changes, the local dynamics of the labeled amino acids in the mutated fibrils could be modified by the introduced perturbations, e.g., due to altered side chain packing in the fibril core. The motional amplitude of the ^13^C–^1^H bond vectors was studied using the DIPSHIFT separated local field experiment, measuring motionally averaged dipolar coupling strengths of any labeled ^13^C–^1^H bond in the peptide backbone or side chain of the fibril. Partial motional averaging of the dipolar coupling strength is conveniently expressed as an order parameter (*S*), which is defined as the ratio of the detected, motionally averaged dipolar coupling and the full dipolar coupling in a completely rigid bond vector [34]. An order parameter of zero, therefore, is obtained for moving C-H bond vectors isotropically, while an order parameter of one indicates the absence of any motional amplitude with a correlation time of less than ~40 µs.

Segmental order parameters for the labeled segments are illustrated in Figure 5 for the backbone (A) and side chain carbons (B). No major differences in the order parameters of the labeled amino acids were observed between WT Aβ_40_ fibrils and the different mutants, proving again that the local structure of the fibrils as the final product of the folding pathway is mostly unaffected by the introduced perturbations.

The order parameters of the backbone carbons were highly rigid in all peptide variants in agreement with a stable cross-β structure for all Aβ fibrils as described in the literature [35]. The rigidity of the Phe_19_ Cβ and Val_21_ Cβ as well as the high flexibility of the phenyl ring and the Val_18_ Cγ/Ala_21_ Cβ in the side chain were retained as well. Noteworthy alterations were only observed in the Phe_20_ mutants. All three Phe_20_ mutants featured an increase in the order parameter of the Phe_19_ Cβ from 0.8 in WT fibrils to 1, indicating full rigidity of these β-carbons. Observed order parameters of *S* > 1 in the Ala_21_ Cα and Phe_19_ Cβ represent experimental errors as there is a theoretical limitation to *S* ≤ 1. It should be noted that dipolar couplings for the Val_18_ Cβ and Ala_21_ Cβ could not be separated. As methyl group rotation averages the order parameter to a minimal value of 0.33 the observed experimental order parameters of ~0.2 for these amino acids indicate additional averaging due to motions of the C–C bond.

### 3.7. Cell Toxicity of the Aβ_40_ Peptides

To investigate the cell toxicity of WT and mutated Aβ_40_ peptides, a standard MTT assay was carried out using the neuronal cell line RN46A. Figure 6 presents the relative cell viability of the variants in comparison to the control group. The cell viability of the control group was set to 100%. RN46A cells were incubated with freshly dissolved lyophilized peptides or with the vehicle for the control group, respectively. Thus, Aβ_40_ peptides were added as unfibrillated monomers/small intermediates.

The results of the MTT assay reveal great differences in the peptide toxicity between WT Aβ_40_ peptides and all tested mutants. As expected, and in agreement with the literature data, the WT peptide was highly toxic and led to a relative cell viability of 39.2 ± 10.3%. All mutations caused a severe decrease or loss of cell toxicity. The Phe_20_ Aβ_40_ mutants were still toxic but to a lesser extent with a relative cell viability of 57.4 ± 7.0% for Phe_20_Lys, followed by 58.9 ± 10.8% for Phe_20_Tyr and 74.0 ± 4.6% for Phe_20_Cha. The variant Gly_33_Ala almost completely lost cell toxicity with a relative cell viability of 99.3 ± 6.8%.

### 3.8. Specific Structural Investigations on the Gly_33_Ala Variant

As the most drastic changes in fibrillation kinetics and cell toxicity occurred in the Gly_33_Ala variant, we investigated an alternatively ^13^C/^15^N-labeled version of the Gly_33_Ala variant with uniformly ^13^C/^15^N-labeled amino acids at position Phe_19_, Asp_23_, Gly_25_, Leu_34_ and uniformly ^15^N-labeled Lys_28_ using solid-state NMR distance measurements. We applied two different preparation protocols. First, we prepared mature Aβ fibrils as in the other experiments and, second, a protocol first introduced by Sarkar et al. [23] where oligomeric structures are prepared and fixed for NMR investigation. In particular, we looked at two important contacts well-described in WT Aβ_40_: the hydrophobic contact between Phe_19_ and Leu_34_ and the salt bridge between Asp_23_ and Lys_28_ [36,37].

The hydrophobic contact between Phe_19_ and Leu_34_ can be well-studied using the two-dimensional ^13^C–^13^C DARR experiment. Figure 7A shows a section of the DARR NMR spectra of fibrils and oligomers grown from the Gly_33_Ala mutant. A long mixing time of 500 ms was applied which allowed for the detection of interresidual contacts. As apparent from the figure, cross peaks between the Phe_19_ ring carbons and the Leu_34_ Cγ, Cδ_1_ and Cδ_2_ indicate magnetization transfer between these carbons, which emerged in the oligomeric as well as fibrillary structures, revealing a proximity of the two amino acids and, therefore, a molecular contact between residues Phe_19_ and Leu_34_ in both states.

The formation of a salt bridge between Asp_23_–Lys_28_ could be precisely examined in ^13^C–^15^N frequency selective REDOR NMR experiments measuring the distance between Asp ^13^COO^−^ and Lys ^15^NH_3_^+^ [14,30]. The results for Gly_33_Ala Aβ fibrils and oligomers are shown in Figure 7B. The ^15^N–^13^C distance could, therefore, be determined to approximately 4.1 Å in fibrils indicating the formation of a stable salt bridge and >7 Å in oligomers indicating that no salt bridge was formed.

## 4. Discussion

The Aβ_40_ peptide is known to form amyloid fibrils. The fibrillation process and the structure of mature fibrils have been shown to be astonishingly robust against local perturbations in the amino acid sequence [11,12,13,14,15,16,17]. This is because the unifying structural motif of all amyloid fibrils, which is the cross-β structure, represents an energetically highly favorable structure [38,39]. In the past, it could be shown that this structure also prevails in most Aβ_40_ peptide mutants [11,12,13,14,15,16,17]. The hydrophobic contact between Phe_19_ and Leu_34_ was found to form very early within the fibrillation process [13]. Rediscovered in various oligomeric structures, this contact seems to represent a first interaction site between hydrophobic domains, which later on results in the formation of the cross-β structure. Interestingly, Aβ_40_ peptides with single mutations of either Phe_19_ or Leu_34_ showed lower toxicity but left the general fibrillar structure largely unchanged. While mutations with altered ring structures like Phe_19_Cha or Phe_19_Nal reduced peptide toxicity [16], less conservative mutations like Phe_19_Gly, Phe_19_Trp or Phe_19_Tyr completely abolished the toxicity of Aβ_40_ [13]. Very conservative mutations of Leu_34_ (to *D*-Leu, Ile or Val) were investigated as well and showed only marginal alterations in toxicity [15]. Previous work led to the assumption that the Phe_19_/Leu_34_ contact, and Phe_19_ especially, harbors fundamental importance for peptide cellular toxicity [40]. This provokes the question: Is the Phe_19_/Leu_34_ contact isolated responsible for the propagation of peptide toxicity or can modifications in the neighboring amino acids cause similar effects? Therefore, we investigated two neighboring sites (Phe_20_ and Gly_33_). This allowed for separating the effects on the peptide’s characteristics between the contact itself and the direct vicinity. In the single mutants, Phe_20_ was replaced by Tyr, Lys or cyclohexylalanine because they showed the most drastic changes in the previous Phe_19_ mutants. Furthermore, Gly_33_ was mutated to Ala to introduce the smallest possible alteration at this site.

To first determine the effect of the mutation on the fibrillation process, we performed fibrillation kinetics measurements. Most of the tested mutants showed a shorter lag time and characteristic time compared to the WT Aβ peptide. The explanation of the partly drastic alteration of the fibrillation kinetics, especially in the Gly_33_Ala mutant, lies in the modification of the folding energy landscape resulting in an acceleration of kinetically relevant steps by destabilizing specific oligomeric structures and/or the emergence of alternative pathways with different oligomers [3,4,41]. The kinetics data suggest an importance of positions Phe_20_ as well as Gly_33_ for fibril growth and the formation and/or stabilization of intermediate species.

In previous studies, the effect of mutations of Phe to Tyr, Lys and Cha was already tested for position Phe_19_. Phe_19_Lys and Phe_19_Tyr showed an elongated lag time and characteristic time compared to WT [11]. In contrast, Phe_20_Lys and Phe_20_Tyr exhibited an acceleration of these kinetic parameters. Phe_19_Cha accelerated fibrillation [16], while Phe_20_Cha slowed it down. Since Phe_19_ points to the hydrophobic interior, mutations to less hydrophobic amino acids seem to slow down fibrillation, while position Phe_20_, pointing outside, seems to profit from more hydrophilic amino acids regarding fibrillation kinetics. It is known that for some proteins such as myoglobin [42] or alpha-lactalbumin [43], a folding process starts with creating localized regions of predominantly hydrophobic residues. It is still unknown how the folding process in Aβ_40_ is directed. The arrangement of hydrophilic and hydrophobic amino acids near Phe_19_/Leu_34_ clearly determines how fast the folding process can proceed.

The TEM results confirm that WT Aβ as well as the Aβ mutants formed fibrils of comparable morphology. X-ray diffraction exhibited a close similarity between the mesoscopic structure of all variants. In more detail, an increased fibril diameter was observed for the Phe_20_ mutants compared to the WT. This suggests slight structural differences between these variants. Two aspects are most likely affected by the mutations:(i)The equatorial reflection in XRD data that corresponds to the distance between the opposing β-strands suggests a slight increase of the intersheet distance but certainly not to an extent to explain the diameter alteration observed in the TEM images. The fact that Phe_20_ according to most Aβ_40_ models [19,44] points to the outside renders a strong influence on this distance unlikely. The moderate increase can be explained by very small packing differences between the opposing β-strands due to alterations in the interaction of the side chains inside the fibril;(ii)Current Aβ_40_ fibril models assume that the fibrils are formed of multiples (usually 2–8) of these opposing β-strands termed protofilaments. The protofilaments line up parallel to each other. The mutations to Phe_20_ may cause alterations of this agglomeration of the protofilaments. In the literature, different diameters for Aβ_40_ WT fibrils are reported depending on preparation protocols. These diameters could be associated with assumed numbers of protofilaments [44,45]. Although it is difficult to assess this further with the data provided by our experiments, we assume that the arrangement of the altered side chains in position 20 affects the alignment of the protofilaments.

Atomistic insight into the local fibril structure near the mutated site was achieved by solid-state NMR. WT and mutated Aβ fibrils featured clearly β-strand-like chemical shifts. Only minor differences between WT Aβ and the other variants occurred. The DIPSHIFT results suggest that the labeled backbone carbons are highly rigid in agreement with the stability of the cross-β structure. All three Phe_20_ mutants resulted in an increased order parameter of the Phe_19_ Cβ carbon compared to the WT. Slight packing differences occurring from the side chain alterations may have caused a constraint that restricted the motion of Phe_19_ Cβ and the location of the phenyl ring while the ring carbons remained flexible. Taken together, the local structure and dynamics near the mutation site Phe_20_ remained largely unaffected by the perturbations. Interestingly, the Gly_33_Ala mutant did not affect experimentally-determined chemical shifts or the flexibility of the labeled residues despite the more restricted dihedral backbone angles of Ala. Thus, Phe_20_ and Gly_33_ amino acids did not exert an important structural influence on Aβ_40_ fibrils.

Although the structure and morphology of the fibrils were only marginally altered, more pronounced effects could be manifested in the toxicity as this depends not on the structure of fibrils but of the oligomeric species. All mutated peptides showed lower toxicity compared to the WT. The Gly_33_Ala mutation completely abolished cell toxicity. Therefore, we investigated the structure of Gly_33_Ala oligomers and fibrils by additional solid-state NMR experiments. We applied a protocol to prepare Aβ oligomers [23] to reveal possible structural differences between oligomers of WT Aβ or the Gly_33_Ala variant. Other NMR-based methods involving pressure jump [46] or antibody binding [47] also represent attractive options to study these transient oligomeric states. The hydrophobic contact between Phe_19_ and Leu_34_ was shown to be preserved in oligomers as well as in fibrils. However, the salt bridge between Asp_23_ and Lys_28_ was only present in fibrils and not in the oligomers.

Our results show that not Phe_19_/Leu_34_ contact alone but also its periphery is crucial for Aβ toxicity. Even with a retained contact between Phe_19_ and Leu_34_, a fibrillation process through non-toxic intermediates can be achieved. Interestingly, mutational studies of Leu_34_ [15] reduced cell toxicity to a much lesser extent then in the Gly_33_Ala variant.

Combining the result from the toxicity assay with the other experiments, certain conclusions arise. Although the structural characteristics of the various fibrils remained largely unchanged, major effects in the fibrillation kinetics occurred. The fact that the final structure of the fibrillation pathway, Aβ fibrils, remained mainly unaffected implies that the perturbations affect mostly transient intermediates resulting in conformational changes on an oligomeric level in agreement with modifications in cell toxicity [48]. Previous work has identified trimers and tetramers as the major species under conditions similar to our toxicity experiments [49]. Another explanation which links the acceleration of the fibrillation process in the Gly_33_Ala mutant to the toxicity results may be the decreased lag time corresponding to a shorter lifetime of the toxic oligomeric species. Studies on Phe_19_ mutated Aβ_40_ peptides with an altered ring structure led to slight reductions of toxicity while no lag time was observable [16]. Major mutations of Phe_19_ extended the lag time and completely abolished toxicity [11,13]. A strong coherence between reduced lag time and reduced toxicity has so far not be observed. However, it is possible that through an altered folding landscape, toxic oligomers are destabilized and more transient.

The observation that the periphery of the Phe_19_/Leu_34_ contact plays a critical role for Aβ_40_ characteristics and toxicity suggests high potential for drug development. The first FDA-approved antibody aducanumab against Aβ is binding to the N terminus of the peptide [50]. While achieving good results in clearing Aβ fibrils from the brain, clinical outcome data remain rather poor [2]. This seems understandable because, from a pathophysiological perspective, the fibrils themselves represent a side product rather than the problem. The current and previous results [13,15,16] suggest that small molecules attacking the region around the Phe_19_/Leu_34_ contact could provide an alternative pharmaceutical strategy to reduce Aβ toxicity. The amino acids we found to be important for that region likely point to the fibril’s exterior and, therefore, are more easily accessible for pharmacological intervention than the Phe_19_/Leu_34_ contact in the fibrillar interior. Phe is a favorable motif for rational drug design since the aromatic ring provides interaction possibilities via various mechanisms such as π stacking, cation–π interactions and C-H–π interactions [51,52,53]. The abolished toxicity in the Gly_33_Ala variant and the reduced toxicity of the Phe_20_ mutants provide hope that it may be possible to engineer small molecules interfering with the folding energy landscape of Aβ_40_ to redirect the fibrillation process via non-toxic intermediates.

On a final note, it should be considered that the individual methods used here required different preparation concentrations and conditions. It is unlikely that fibrillation started from true Aβ monomers when incubated at higher concentrations. Monomeric conditions for Aβ_40_ arise only when the peptide is diluted to very low concentrations (≤150 nM, [54]).

## Figures and Tables

**Figure 1 biomolecules-11-01848-f001:**
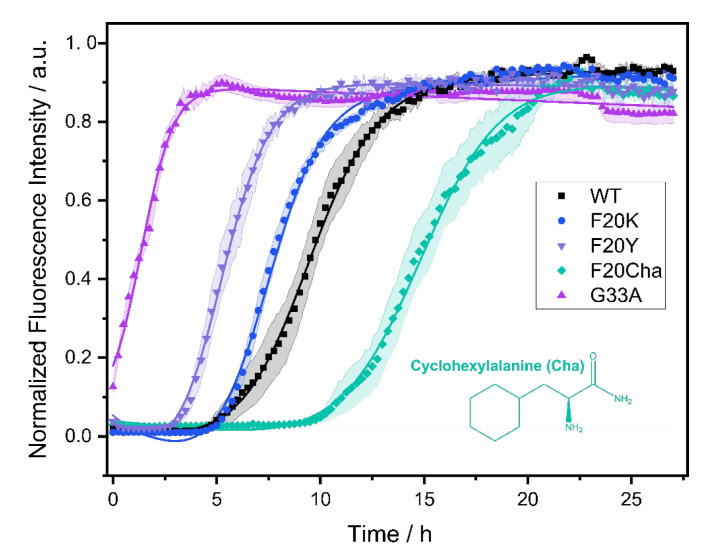
Fibrillation kinetics of Aβ_40_ WT and the respective mutants recorded by the standard Thioflavin T (ThT) fluorescence assay. Solid lines represent best fit simulations according to equation. (1). The shaded areas represent the standard error of the mean. Intensity data measurements were carried out in three independent experiments in quintuplicate for each peptide. Experimental errors in the determination of the lag time and fibrillation time were calculated from fitting errors.

**Figure 2 biomolecules-11-01848-f002:**
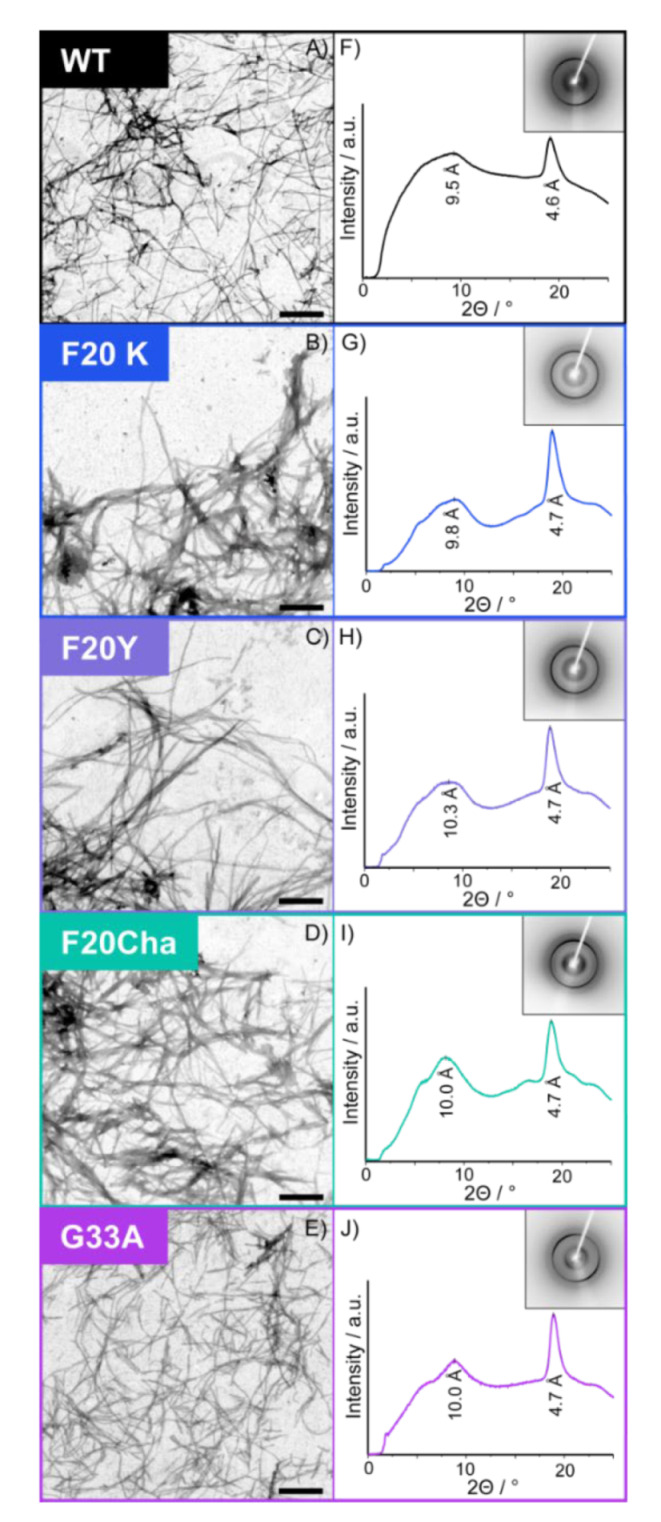
Transmission electron micrographs (**A**–**E**) and X-ray diffraction patterns (**F**–**J**) of fibrils of the Aβ_40_ variants. The scale bar represents 400 nm.

**Figure 3 biomolecules-11-01848-f003:**
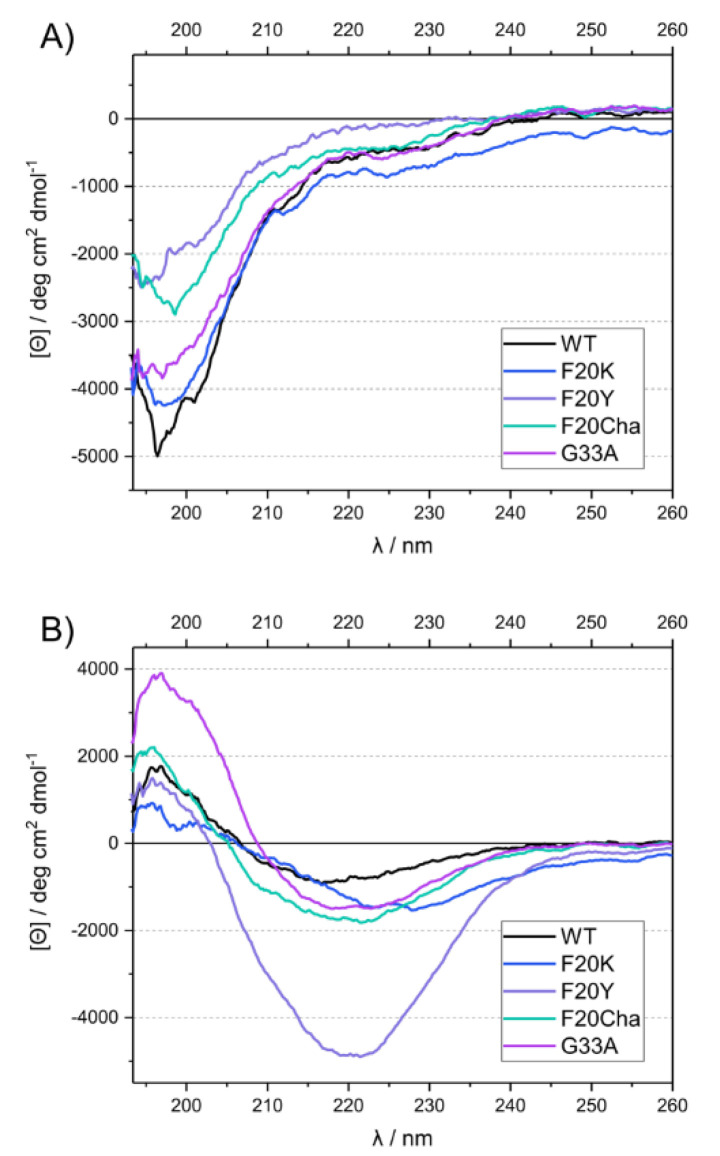
UV circular dichroism spectra of WT Aβ_40_ and mutant Aβ_40_ peptides before (**A**) and after incubation (**B**).

**Figure 4 biomolecules-11-01848-f004:**
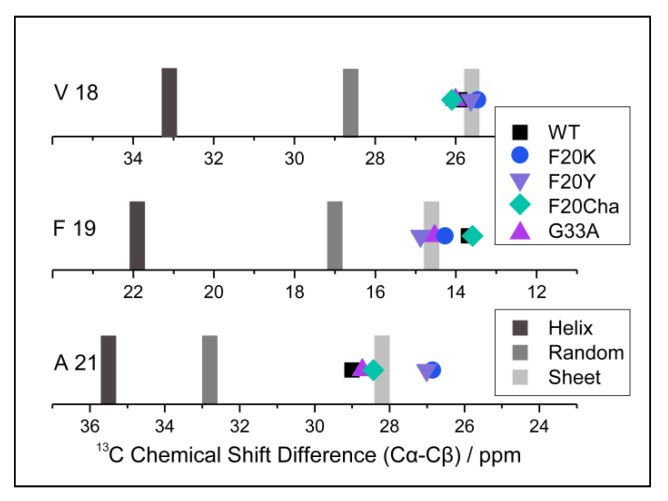
^13^C NMR Cα–Cβ chemical shift differences of the Aβ_40_ fibril variants (symbols) and reference values for α-helix, random coil and β-sheet structure (gray bars) for the labeled amino acids Val_18_, Phe_19_ and Ala_21_. ^13^Cα–^13^Cβ chemical shift differences represent a measure of the local secondary structure, which is independent of external chemical shift referencing.

**Figure 5 biomolecules-11-01848-f005:**
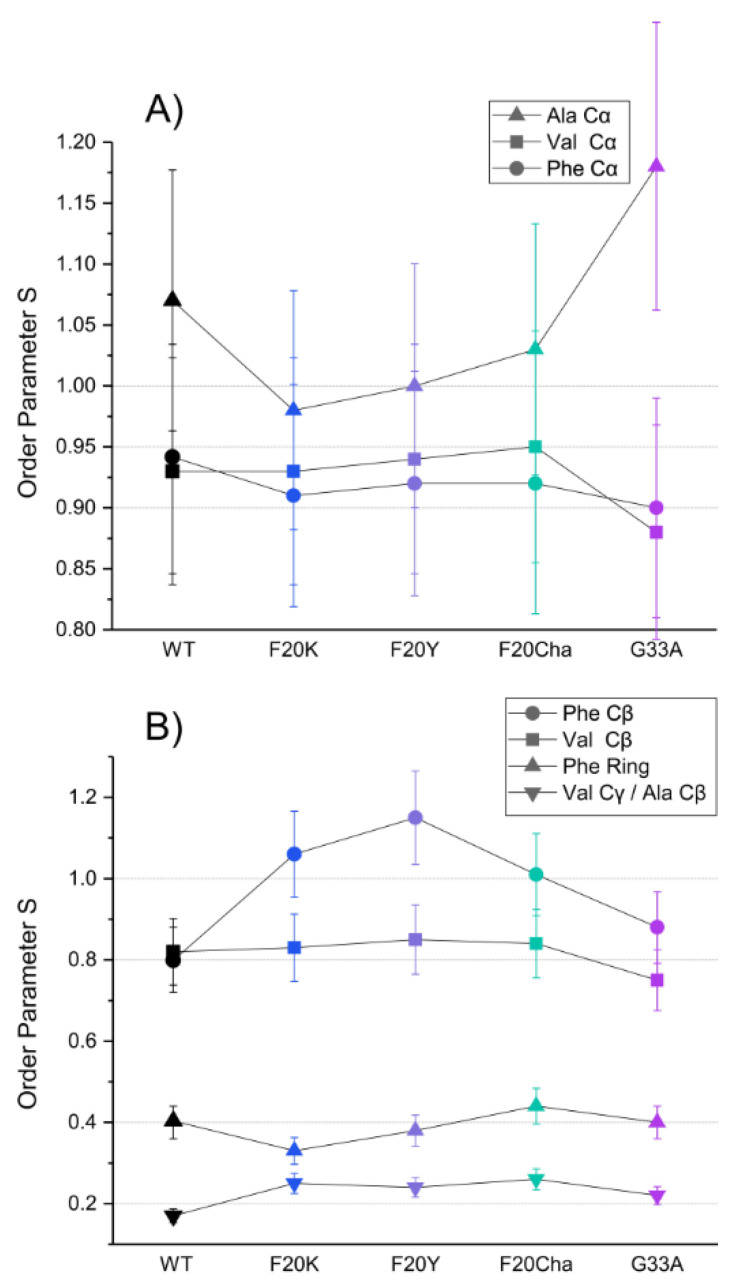
DIPSHIFT derived ^13^C–^1^H order parameters for the labeled ^13^C atoms of Val_18_, Phe_19_ and Ala_21_ in amyloid fibrils of the Aβ_40_ peptide variants as representation of the segmental molecular dynamics. (**A**) backbone sites, (**B**) side chains. Error bars were estimated as ±10% of the order parameter value. The order parameters for the Val_18_ Cβ and Ala_21_ Cβ could not be resolved.

**Figure 6 biomolecules-11-01848-f006:**
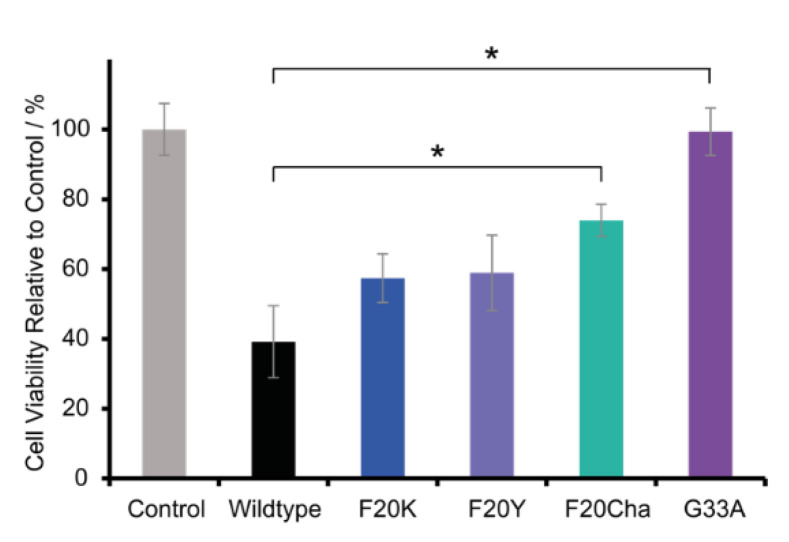
Relative cell viability of RN46A cells incubated with peptide variants or the vehicle for the control group, respectively. The concentration of Aβ (1–40) WT or the mutant species was 100 µM, cells were incubated for 60 h at 37 °C. The cell viability of the control group was set to 100%. Error bars represent standard errors of the mean, * represents *p* < 0.005.

**Figure 7 biomolecules-11-01848-f007:**
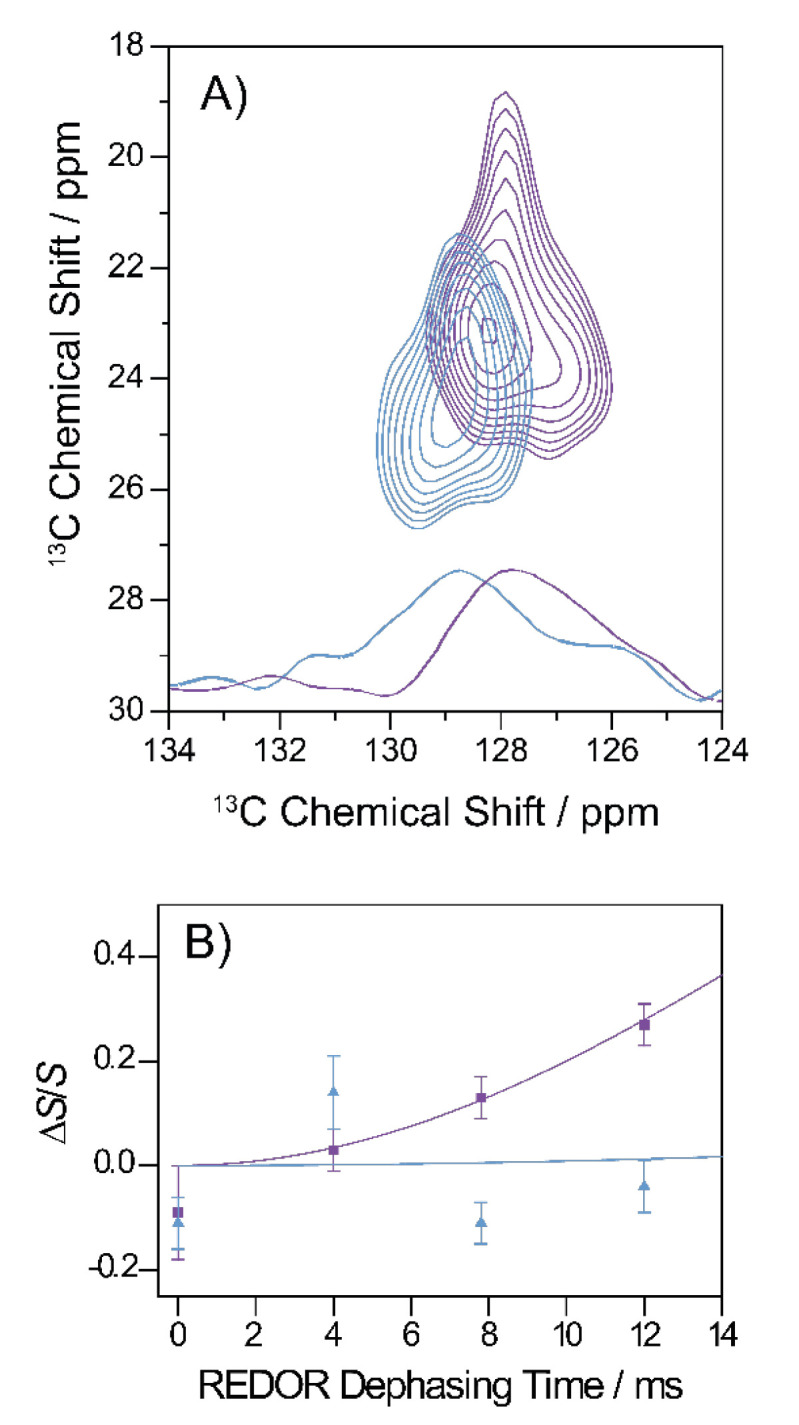
Summary of the specific tertiary contact analyses on Gly_33_Ala oligomers and fibrils. (**A**) Segment of the ^13^C-^13^C DARR correlation spectra (500 ms mixing time) of oligomers (blue) and fibrils (violet). The cross peaks indicate a molecular contact between Phe_19_ and Leu_34_ in oligomeric and fibrillary state. Rows extracted from the 2D contour maps (at 23.1 ppm for fibrils in violet and at 24.5 ppm for the oligomers in blue) are plotted on the bottom of the graph. (**B**) REDOR dephasing curve of the Gly_33_Ala fibrils (violet) and oligomers (blue). The Asp_23_-^13^COO^−^-Lys_28_-^15^NH_3_^+^ distance in the side chains of fibrils could be determined to 4.1 Å, indicating the formation of a stable salt bridge. In oligomers, no REDOR dephasing was observed between the side chains of Asp_23_ and Lys_28_ indicating the absence of the salt bridge.

## Data Availability

All raw data of this publication is available from the authors upon request: daniel.huster@medizin.uni-leipzig.de.

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
