# Peer review of "Probing the Influence of Single-Site Mutations in the Central Cross-β Region of Amyloid β (1–40) Peptides"

_biomolecules, 2021, doi:10.3390/biom11121848_

Round 1

Reviewer 1 Report

It is important to understand the aggregation properties of amyloid-beta peptides that are related to Alzheimer's disease. This manuscript reports in an in-vitro investigation of abeta fibrils formed by the modified peptides (with Phe20 mutated to Lys, Tyr or the non-proteinogenic cyclohexyl-25 alanine or Gly33 mutated to Ala) using a variety of biophysical techniques as well as toxicity to neuronal cells. Overall, this is a well carried out study and the reported results are interesting. It can be published after revision. Suggestions for revision are given below. 

1) It is useful to include the error range/bars for the ThT data in Figure 1. Mention if the reported results are based on triplicates. 

2) The dock-and-lock mechanism of abeta aggregation and fibril formation. This is well investigated via theoretical & simulations by Straub and Thirumalai (J Phys Chem B. 2009 Oct 29;113(43):14421-30) and the recent NMR experimental results showing docking of monomer/oligomers to abeta-fiber-seeds (Brender et al ChemComm (2019)) can be included. 

3) A recent study by Ad Bax et al on the oligomeric structure of abeta and the use of pressure can be mentioned. In addition, a lowly populated disordered 3-10 helical structure of abeta (as an intermediate) by NMR reported in BBRC (2011) shows the importance of Phe19 residue. This can be mentioned. In fact, amyloid inhibitors have been shown to interact with the region of abeta identified in this study (see the work of Mi Hee Lim and coworkers). 

4) Studies by Gazit and coworkers have shown the roles of pi-pi interaction in the aggregation of abeta and other amyloid peptides. This can be mentioned. 

5) It would be helpful to readers to show 1D slices taken from Fig.6 for comparison (include them in SI). 

6) It would be useful to include the following very recently reported articles related to the reported study. 

  • Fatafta et al "Disorder-to-order transition of the amyloid-β peptide upon lipid binding" (2021)
  • Nguyen et al Chem.Rev.(2021)
  • Cawood et al "Visualizing and trapping transient oligomers in amyloid assembly pathways" (2021) 
  • Sharma et al "Kinetics theories to understand the mechanism of aggregation of protein..." (2021).
  •  

Minor changes: 

"Developing small molecules or antibodies that alter or inhibit this fibrillation process satisfactorily has largely failed in the last decade."

This is true, but it should be modified to indicate that the fibrillation process occurring in solution. 

Author Response

It is important to understand the aggregation properties of amyloid-beta peptides that are related to Alzheimer's disease. This manuscript reports in an in-vitro investigation of abeta fibrils formed by the modified peptides (with Phe20 mutated to Lys, Tyr or the non-proteinogenic cyclohexyl-25 alanine or Gly33 mutated to Ala) using a variety of biophysical techniques as well as toxicity to neuronal cells. Overall, this is a well carried out study and the reported results are interesting. It can be published after revision. Suggestions for revision are given below. 

Response: We thank the reviewer for her/his encouraging remarks.

1) It is useful to include the error range/bars for the ThT data in Figure 1. Mention if the reported results are based on triplicates. 

Response: We modified the plot shown in Fig. 1 according to the reviewer’s recommendation.

2) The dock-and-lock mechanism of abeta aggregation and fibril formation. This is well investigated via theoretical & simulations by Straub and Thirumalai (J Phys Chem B. 2009 Oct 29;113(43):14421-30) and the recent NMR experimental results showing docking of monomer/oligomers to abeta-fiber-seeds (Brender et al ChemComm (2019)) can be included. 

Response: We included these citations.

3) A recent study by Ad Bax et al on the oligomeric structure of abeta and the use of pressure can be mentioned. In addition, a lowly populated disordered 3-10 helical structure of abeta (as an intermediate) by NMR reported in BBRC (2011) shows the importance of Phe19 residue. This can be mentioned. In fact, amyloid inhibitors have been shown to interact with the region of abeta identified in this study (see the work of Mi Hee Lim and coworkers). 

Response: We have added this important information and thank the reviewer for pointing it out.

4) Studies by Gazit and coworkers have shown the roles of pi-pi interaction in the aggregation of abeta and other amyloid peptides. This can be mentioned. 

Response: Thank you for this suggestion, this has been mentioned now in the revised paper.

5) It would be helpful to readers to show 1D slices taken from Fig.6 for comparison (include them in SI). 

Response: We have added these slices to Fig. 6 (since we do not have SI with the paper).

6) It would be useful to include the following very recently reported articles related to the reported study. 

  • Fatafta et al "Disorder-to-order transition of the amyloid-β peptide upon lipid binding" (2021)
  • Nguyen et al Chem.Rev.(2021)
  • Cawood et al "Visualizing and trapping transient oligomers in amyloid assembly pathways" (2021) 
  • Sharma et al "Kinetics theories to understand the mechanism of aggregation of protein..." (2021).

Response: Done

Minor changes: 

"Developing small molecules or antibodies that alter or inhibit this fibrillation process satisfactorily has largely failed in the last decade."

This is true, but it should be modified to indicate that the fibrillation process occurring in solution. 

Response: The statement has been rewritten to read:

Developing small molecules or antibodies that alter or inhibit this fibrillation process in solution has largely failed in the last decade.

Reviewer 2 Report

The manuscript from Fritzsch and collegues focuses on the study of amyloidogenic properties of peptides obtained mutating Abeta1-40 peptide involved in Alzheimer’s disease in positions 20 (Phe20) and 33 (Gly33) that are neighbouring residues of Phe19 and Leu34 recently revealed as crucially involved in toxicity mechanisms resulting from cell interaction with Abeta peptide aggregates. In this study the correlation structure-toxicity of the species produced by the Abeta mutants’ aggregation is investigated, too.

The study is well conducted and data are clearly described. However, there are some points that necessary should be addressed and that are reported below

Major points:

  • In the peptide preparation, no check on the initial species is carried out. Since the preparation procedures reported are different depending the technique to be used (as an example DMSO for dissolving sample was used for ThT fibrillogenesis experiment and not for CD measurements) a control on the initial species to confirm the prevalence of monomeric species (for example HPLC analysis) should be reported. This is especially true, since the sample seems to be never filtered before starting fibrillogenesis. How are the authors sure to start from monomeric species as specified in certain manuscript points?
  • As known from literature, the most toxic amyloid species are considered the intermediate amyloid-β (Aβ) oligomeric aggregates, like the low-molecular-weight oligomers (dimers, trimers..). In the contrast mature fibrils seem to be harmless. Therefore, since the authors are sure to add unfibrillated intermediates species to cells, they justify the toxic action exerted by the wildt ype sample as found by MTT assay. However, Abeta species remain for 60 hours at 37°C before controlling toxicity. Are do the authors sure that the sample does not evolve during this time? The evolution could be significant (even if not an aggregation “on pathway” because, in this case, it is without agitation) if samples are at the concentration of 100microM. Probably this concentration reported in material and methods (MTT section) is the initial one, before dilution into cells. What is the final? Can the authors control the oligomeric/structural properties of the species resulting at that concentration when Abeta peptide is kept for 60 hours at 37°C (for example by CD)? This could be important, in general, for addressing the structure/function relationship issue concerning amyloids.
  • In some cases, also plateau value of ThT fluorescence emission could give important information because it is correlated to the total amount of fibrils formed during the process. Is it possible to report the fibrillogenesis kinetics assessed by ThT assay (Fig. 1) also with unnormalized values in order to compare the plateau values for the different samples?

Minor points:

  • 8. Instead of figure 3 legend the figure 2 legend is reported.
  • Errors in figure numbers are reported (as an example there are two legends with “figure 5” and in figure 7 legend is called “figure 6)

Reviewer 3 Report

In this manuscript, the authors describe the effect on fibrillation of some mutations points introduced in the vicinity of the significant hydrophobic contact between Phe19 and Leu34 of Ab40.
They characterize and compare the amyloid formation of four different mutants (F20KF20YF20Cha and G33A) using different biophysical techniques (ThT fluorescence, TEMXray diffraction, CD, ssNMR) and cell viability assays. The experiments are clearly undertaken and the results are supported with a good discussion part describing the structure - toxicity relationship of amyloid beta peptide and providing important information in the amyloid field. The results presented in this work could help in the design of potential therapeutic strategies.
For such reasons, I recommend publication of this manuscript in Biomolecules.

Minor points:

- Text legend of Figure 3 (CD spectra) is not correct.
- Line 334: write « dihedral angles » instead of « dihedral angels »
- Line 562: a point « . » is missing after « drug development ».

Author Response

In this manuscript, the authors describe the effect on fibrillation of some mutations points introduced in the vicinity of the significant hydrophobic contact between Phe19 and Leu34 of Ab40.
They characterize and compare the amyloid formation of four different mutants (F20K, F20Y, F20Cha and G33A) using different biophysical techniques (ThT fluorescence, TEM, Xray diffraction, CD, ssNMR) and cell viability assays. The experiments are clearly undertaken and the results are supported with a good discussion part describing the structure - toxicity relationship of amyloid beta peptide and providing important information in the amyloid field. The results presented in this work could help in the design of potential therapeutic strategies.
For such reasons, I recommend publication of this manuscript in Biomolecules.

Response: We thank the reviewer for her/his positive evaluation of our work.

Minor points:

- Text legend of Figure 3 (CD spectra) is not correct.

Response: We apologize and have corrected the legend to Fig. 3 (it happened during transfer of the text into the Biomolecules template)

- Line 334: write « dihedral angles » instead of « dihedral angels »

Response: Corrected

- Line 562: a point « . » is missing after « drug development ».

Response: Corrected

Round 2

Reviewer 2 Report

But we do not claim that we start from monomeric species. We added some explanation about this point to the discussion section.

The revised version of the manuscript is improved and the raised issues were point addressed. I also agree with the importance of comparing Ab WT and mutants at the same concentration and preparation conditions for each technique. Therefore, it is important that the authors changed the sentence (not indicated in yellow in the new version but present in CD materials and methods section old manuscript version) in which they indicated with the term “monomers” the initial sample.

The toxicity is indeed tested at 100 μM final concentration. At this concentration, it is possible that fibrillar species emerge at least at the later stages of incubation (depending on the lag time of fibril formation). We note that all the species, including the WT, are tested the same way. Given that there is very little difference between their end-states, it is compelling to infer that it is the difference in the intermediate states which matter for toxicity.

Could the authors confirm this hypothesis, for example by HPLC? This could be interesting to add information on general Ab amyloid structure-toxicity correlation.  

Author Response

Our previous response: The toxicity is indeed tested at 100 μM final concentration. At this concentration, it is possible that fibrillar species emerge at least at the later stages of incubation (depending on the lag time of fibril formation). We note that all the species, including the WT, are tested the same way. Given that there is very little difference between their end-states, it is compelling to infer that it is the difference in the intermediate states which matter for toxicity.

Reviewer’s remark: Could the authors confirm this hypothesis, for example by HPLC? This could be interesting to add information on general Ab amyloid structure-toxicity correlation. 

Our response: We agree that it is exciting to understand which oligomers are the most toxic ones. Work from S. Maiti’s laboratory has worked on that issue (Chandra et al., Biophys. J. 113 (2017) 805). Using FCS and single molecule photobleaching, it was found that the average hydrodynamic radios is about 1.6 nm, which would correspond to trimers and tetramers. We have added a sentence and a reference to the manuscript.